# In Vitro and Ex Vivo Evaluation of *Mangifera indica* L. Extract-Loaded Green Nanoparticles in Topical Emulsion against Oxidative Stress and Aging

**DOI:** 10.3390/biomedicines10092266

**Published:** 2022-09-13

**Authors:** Zaheer Ullah Khan, Taous Khan, Abdul Mannan, Atif Ali, Jiang Ni

**Affiliations:** 1Department of Pharmacy, Abbottabad Campus, COMSATS University Islamabad, Abbottabad 22010, Pakistan; 2Department of Pharmacy, Affiliated Hospital of Jiangnan University, Wuxi 214000, China

**Keywords:** *Mangifera indica* L., nanolipid carriers, antioxidant, oxidative stress, antiaging, skin permeation

## Abstract

Although *Mangifera indica* L. extract (M-Ext) of the peel and kernel possesses potent antioxidant and excellent antiaging qualities, the effects are only partially seen because of the skin’s limited ability to absorb it. M-Ext was loaded into nanolipid carriers (M-NLCs) in this work to create a green topical formulation that would boost antiaging efficacy and address penetration deficit. Compound identification was done using GCMS and atomic absorption spectroscopy for heavy metals in M-Ext. M-Ext was also evaluated against oxidative stress antioxidant enzymes. The M-NLCs were fabricated and evaluated for their physicochemical characterizations. Cytotoxicity and cell permeation analysis of M-Ext and M-NLCs were carried out in fibroblasts and HaCaT cell lines. An ex vivo permeation study of M-Ext and M-NLC-loaded emulsion was performed through rat skin and the kinetic parameters were determined. Kinetic data showed that the ex vivo permeation of M-NLC-loaded emulsion through rat skin followed the Higuchi model. The safety profile was evaluated in human volunteers after written consent. Three months’ in vivo investigations were conducted using the optimized M-NLC-loaded emulsion and vehicle (B-NLC-loaded emulsion) on human cheeks for comparison. The volunteers’ skin erythema level, melanin contents, TEWL index, moisture contents, sebum level, elasticity, pH, and pore size were examined after the first, second, and third month via noninvasive techniques. There were significant findings for physicochemical characterizations and in vitro and ex vivo studies. The findings demonstrate that the green nanolipid carriers amplified the overall antioxidant effectiveness and may represent an emerging treatment strategy for oxidative stresses and aging.

## 1. Introduction

Researchers have recently turned their attention to natural products as a result of the rising demand for natural green formulations and cosmetic items. With the notion of better safety and fewer adverse effects compared to their synthetic equivalents, their use is increasing constantly across the world [1]. In 2012, the global market for herbal cosmetics was $1500 billion and estimated to grow at a rate of 25% per annum in both developed and developing countries [2]. The natural active ingredients used in these products have strong antioxidants, vitamins and other bioactive compounds that slow the skin aging process [1,3].

*Mangifera indica* L. is a popular fruit whose extract has been reported to have several beneficial medicinal properties other than nutritional purposes [4]. Its peels and kernel extracts comprising 40–50% of its weight contain a high amount of bioactive compounds i.e., gallic acids, quercetin, mangiferin, catechin, epicatechin, quercetin, isoquercetin, fisetin, rutin, ellagic acid, gallic acid, gallates, condensed tannins, gallotannins, anthocyanin, and astragalin [5,6,7,8,9]. These compounds have good potency to control the overproduction of FRs [10]. *M. indica* crude extract (peels and kernel) showed tyrosinase inhibition [11,12] and reversible inhibition of elastase and collagenase enzymes [13], which makes it appropriate for use in pharmaceutical and cosmetic products to treat dermatitis, psoriasis, acne, and premature skin aging [14]. The topical application of *M. indica* kernel extract is limited due to its low shelf life, poor solubility, and low skin permeation [15].

Topical nanolipid carriers (NLCs) have the potency to address the limitations related with conventional topical formulations loaded with natural extracts via such physicochemical properties as their controllable and ultrasmall size, more surface area, functionalizable structure, and increased stability [14]. Such delivery systems amend the bioactive compound to perform longer and more effectively within the body [16]. Topical nanolipid carriers are lipid-based vehicles considered the most suitable topical delivery system to enhance the efficacy of treatment regimens for skin aging [17,18,19]. They can effectively enhance the aqueous solubility, skin permeability and stability of natural extracts during shelf life in topical formulations [20].

In this study, we extracted *M. indica* peels and kernel (M-Ext) and developed a green formulation of M-Ext-loaded NLCs. The green formulation was characterized for physicochemical characteristics and evaluated for cytotoxicity and permeation. The antiaging efficacy of the developed formulation was evaluated in human volunteers for three months via noninvasive techniques. The development of the NLC-based green formulation aimed (i) to protect the bioactive compounds that might be prone to degradation; (ii) to enhance the skin penetration and augment bioavailability of M-Ext; (iii) to prevent the interference with the enzymatic and antioxidant activities of the extract; and (iv) to permit the fabrication of a robust NLC-based green product that might be used topically to treat and fend off oxidative skin problems and early skin aging.

## 2. Materials and Methods

### 2.1. Materials

Dimethyl sulfoxide (DMSO), Dist. H_2_O, hydrochloric acid (HCl), sodium hydroxide (NaOH), DPPH, gallic acid (GA), ascorbic acid (AA), AlCl_3_, sodium phosphate, FeCl_3_, potassium acetate, phosphate buffer, MeOH, cetyl alcohol and rhamnolipid 90 were bought from Sigma-Aldrich (St. Louis, MO, USA). Folin–Ciocalteu reagent (FCr) and sodium carbonate were bought from Merck (Darmstadt, Germany). All other chemicals used were of analytical grade and used as received.

### 2.2. Methods

#### 2.2.1. Preparation of *Mangifera indica* L. Kernel and Peel Extract

Pure methanol (10:1, *v*/*w*) was used for the extraction of mango waste (peels and kernel) dried fine powder via stirring. After stirring, the extract was purified and solvent evaporation was carried out via rotary evaporator. The extract was dried and kept at a −4 °C. The percentage yield was determined by Equation (1).
**Yield (W %) = W1/Wo × 100**(1)
where W is the yield (%), Wo is the weight of mango peels and kernel extract (M-Ext), and W1 is the dried extract weight (g).

#### 2.2.2. GC-MS Analysis

GC-MS analysis was performed to analyze and screen compounds in the M-Ext of peels and kernel of *M. indica* L. A standard method was used to run GC-MS. Identification was based on the molecular structure, molecular mass and calculated fragments. Interpretation of mass-spectrum GC-MS was conducted using the database of National Institute Standard and Technology (NIST) library (2005), which has more than 62,000 patterns.

#### 2.2.3. Atomic Absorption Spectroscopy

Metal concentration determination is of prime importance as they play an active role in nutrient deficiency, treatment and toxicity [21,22]. Atomic absorption spectrometer AAnalyst (PerkinElmer, Waltham, MA, USA) was used for evaluation of different and metal concentration. The M-Ext were prepared for minerals and metal analysis, 0.5 g of extract was mixed with conc. HNO_3_:HCl (3:1) mixture and heated at 85 °C for 3 h. Then, HClO_4_ 1 mL was added to increase the oxidation process in digestion. After that, solution was filtered and distilled water added to make the solution 50 mL [23].

#### 2.2.4. Effect on Antioxidant Enzymes

##### Cells and Culture Medium

Peritoneal macrophages were isolated from mice using 1% thioglycolate media. These cells were seeded in a 96-well plate and maintained in the incubator (at 95% air and 5% CO_2_ humidified environment at temperature of 37 °C) as reported previously. The cells were suspended in DMEM supplemented with 10% FBS (fetal bovine serum) and 1% combination of antibiotics (100 μg/mL of streptomycin and 100 U/mL of penicillin). The samples were dissolved in 0.2% dimethyl sulfoxide (DMSO) [24].

##### Effect of Extract on Antioxidant Enzymes GSH, GST, Superoxide Dismutase (SOD), POD, and Catalase

The effect of M-Ext on oxidative stress markers, such as GSH, GST, catalase, SOD, and POD in the media of the peritoneal macrophages stimulated with the lipopolysaccharide (LPS) was determined. GSH was determined by adding the cDNB and culture media as reported previously [24,25]. The concentration of the GSH was determined at 340 nm wavelength using an ELISA plate reader as reported, and similarly the concentration of the GST using dTNB, PBS and culture media as reported previously. The readings were noted at 350 nm using the ELISA plate reader. The concentration of the catalase was determined by adding 0.1 mL of the homogenate with 2.9 mL of the H_2_O_2_ buffer and the absorbance was recorded at 240 nm [24,25]. The SOD concentration was assessed by mixing Tris-EDTA (50 mM and pH 8.5) and pyrogallol (24 mM) and the absorbance was noted at 420 nm. The POD concentration was determined as reported previously [26].

#### 2.2.5. Total Phenolic Contents (TPC)

The Folin–Ciocalteu reagent (FC) technique was used to evaluate the phenolic contents (PC) in M-Ext with slight modifications [27]. Briefly, separate stock solutions of the plant and gallic acid and their fractions were prepared. Then, a volume of 20 µL from each fraction and 90 μL FC reagent was shifted into the wells and incubated for 5 min at 37 °C. Later, 90 μL of 7% *w*/*v* NaCO_3_ solution in distilled water was added and incubated for 1 h. The absorbance was noted at 765 nm via microplate reader (BioTek, Winooski, VT, USA). Gallic acid was used to render the results as equivalents per mg of the dry weight of M-Ext.

#### 2.2.6. Total Flavonoid Contents (TFC)

TFC in M-Ext were found by chlorimetric assay with slight changes [27]. Briefly, 5.0 mg of extract was dissolved in 5.0 mL of methanol and fractions were made. A 20.0 μL volume from fraction solution was shifted into the 96-well microplate, followed by the addition of 10 μL 10% aluminum chloride, 10 μL 1 M C_2_H_3_O_2_K solution, and 160 μL distilled water to bring the total volume up to 200 μL and then incubated for 30 min at 25 ± 5 °C. The absorbance was noted at 415 nm using a microplate reader (BioTek, Winooski, VT, USA). Results were reported as milligram of quercetin equivalents per g of M-Ext dry weight.

#### 2.2.7. 2,2-Diphenyl-1-Picrylhydrazyl Scavenging Assay

The antioxidant activity of the M-Ext was evaluated by DPPH assay [28]: 100 µM of DPPH solution was prepared by dissolving 1 mg DPPH in 25 mL of 10% DMSO [29], 10 µL from various fractions of M-Ext (1–5 μg/mL) and 90 µL from DPPH solution were mixed in the 96-well plate, and concentration was made to 500 µg/mL. After that, the mixture was incubated at 37 °C for 30 min. The decrease in absorbance was noted at 517 nm by microplate reader (BioTek, Winooski, VT, USA). The reduction indicates FRs scavenging activity, given by Equation (2).
(2)$$Inhibition\;\left(\%\right)=\frac{\left(Abs\;of\;negative\;control-Abs\;of\;sample\;solution\right)}{\left(Abs\;of\;negative\;control\right)}\times 100$$

#### 2.2.8. Preparation of NLCs

A modified high shear homogenization technique was applied for the development of NLC [30]. Briefly, 400 to 600 mg of lipids blend of cetyl alcohol and oleic acid was completely dissolved in ethanol along with 5 mg of soya lecithin and extract (10–20 mg), forming organic phase. Organic phase was heated below boiling point of ethanol 78.3 °C. 25 mL water solution containing 400 to 800 mg rhamnolipid (Rh) 90 forming aqueous phase was prepared and heated to above the melting point of cetyl alcohol. Then organic-phase was mixed drop-wise to aqueous phase with constant homogenization at 10,000 rpm. The prepared dispersion was then bath sonicated and cooled to room temperature, containing M-NLC [31,32]. In production of B-NLC extract was not added to the organic-phase. Mannitol 3.0% was added as a lyoprotectant to avoid damage and prevent particle size elevation [33].

#### 2.2.9. Particle Size (PS), Zetapotential (ZP), and Polydispersity Index (PDI)

NLCs were diluted with double distilled H_2_O (×20). The PS, PDI, and ZP were assessed at 25 °C through dynamic light scattering (DLS) using a NanoZS90 (Malvern Instruments Ltd., Malvern, UK) [33].

#### 2.2.10. Entrapped Phenolic Contents

A weighed amount of M-NLCs was liquified in methanol. After half an hour’s centrifugation at 5000 rpm for 15 min, the supernatant was passed through a 0.22 μm filter membrane and analyzed at 765 nm (λ_max_); the concentration of phenolic compounds was measured [34]. EE was computed via Equation (3):**EE % = [Tpc-S/Tpc] × 100**(3)
where T_pc_ = total quantity of PC, S = quantity of PC in the supernatant, and T_pc_-S = quantity of PC entrapped.

#### 2.2.11. Fourier-Transform Infrared Spectroscopy (FTIR)

To certify the entrapment of M-Ext in nanocarriers and interaction between B-NLCs ingredients and extract, FTIR spectroscopy was accomplished using FTIR spectroscopy (Tensor 27, Bruker, Billerica, MA, USA) over the 4000–6000 cm^−1^ of scanning wavelength [33].

#### 2.2.12. Morphological Observation

The morphological observation of M-NLCs were accomplished by transmission electron microscopy (JEM-1200EX, JEOL, Tokyo, Japan) [35].

#### 2.2.13. Cytotoxicity Studies

To evaluate the cytotoxicity, standard MTT assay was used. Human normal keratinocytes (HaCaT) and fibroblast cell lines were used for cytotoxicity studies. HaCaT were cultivated in DMEM and fibroblast in RPMI 1640. All cultures were complemented with FBS, streptomycin, and penicillin in a humidified incubator with 5% CO_2_ at 37 °C [36]. After incubation, the media were aspirated, and each well supplied with fresh medium containing samples of different concentrations and further incubated for a specified time. Next the MTT solution was mixed and incubated up to 4 h, then DMSO was added and gently shaken for 5–15 min. Absorbance was noted at 570 nm and cell viability was determined via Equation (4) [37].
(4)$$Cell\;viability\;\left(\%\right)=\frac{\mathrm{OD}\;\mathrm{values}\;\mathrm{of}\;\mathrm{dosing}\;\mathrm{group}-\mathrm{blank}\;}{\mathrm{OD}\;\mathrm{values}\;\mathrm{of}\;\mathrm{control}\;\mathrm{group}-\mathrm{blank}}\;\times \;100$$

#### 2.2.14. Cell Permeation Studies

To evaluate the permeation of M-NLCs and M-Ext across HaCaT cells, they were cultivated in 24-wells transwell inserts at a concentration of 1 × 10^5^/cm^2^. After 17–21 days, the transepithelial electrical resistance (TEER) was assessed by EVOM ohmmeter to evaluate the cell’s monolayer integrity at TEER values 150 to 300 Ω·cm^2^. Cells were rinsed away thrice, then M-Ext-E and M-NLCs were supplemented at amount of 60 μg in 0.2 mL (300 μL/mL) HBSS (Hank’s balanced salt solution) to the upper chamber of the transwell, while to the lower chamber 1.0 mL HBBS was added. Cells were kept at 37 °C with constant orbital shaking at 50–60 rpm. At different intervals (1–6 h), 500 μL HBSS was withdrawn, and the same amount of fresh HBSS was supplemented into the lower chamber. The samples were lyophilized and liquefied in ACN and PC were quantified [34]. The cumulative transport mass (Q, µg) was computed via Equation (5):**Q = Ci × V + ∑ Ci − 1 × 0.5**(5)
where, V is the mass of the solution in the lower chamber of transwells (mL) and Ci is the PC concentration (µg/mL) [35,38].

#### 2.2.15. Preparation of M-NLC-Loaded Emulsion

Two formulations of O/W emulsion (data not shown) were prepared containing 2% M-Ext. In one formulation, M-NLCs were added till 2% concentration of M-Ext was achieved while in another M-Ext was loaded.

#### 2.2.16. Ex Vivo Diffusion Studies

For ex vivo diffusion studies of M-Ext and M-NLC-loaded O/W emulsion the receptor compartment of Franz’s diffusion cell was filled with 11 mL of PBS keeping 5.5 pH, and the donor chamber filled with 1 g of formulation, i.e., M-Ext loaded emulsion (M-Ext-E), and M-NLC-loaded emulsion (M-NLC-E) respectively. Shaved and clean rat skin was used as a membrane. Formulations were covered by parafilm to avoid evaporation. The solution in receptor chamber was stimulated at 210 rpm at 37 °C. 1 mL of PBS withdrawn at 0.5, 1.0, 2.0, 4.0, 8.0, 16.0, 24.0, 36.0 and 48.0 h. Fresh phosphate buffer solution was further added to the receptor chamber to keep the volume constant during experiments. The samples were evaluated using 1800 UV-visible spectrophotometer (Shimadzu, Kyoto, Japan) at 765 nm to measure the phenolic contents [34].

#### 2.2.17. Rheological Studies

The viscosity of M-NLC-E was determined using a rheometer (RM_200_ Touch, Lamy Rheology, Champagne au Mont d’Or, France). A measuring-system twelve of spindle R-III was utilized. The viscosity of the formulation was measured at a speed of 100 RPM for 60 s at 25 ± 0.5 °C. Measures were as viscosity = f_time_. The viscosity (f_time_) allows measuring at an immobile shear rate throughout a determined time [39]. Measuring protocols were pre-shearing time; 5 s, pre-shearing rate: 10 S 1; shear rate: 50 S 1; time: 60 s; and an immediate start.

#### 2.2.18. pH of M-NLCs at Different Storage Conditions

The information about the pH of formulation is important and is considered a quality criterion [40]. M-NLC-E pH was measured using a pH meter (Hanna, Italy) via a direct method. Readings were performed in triplicates for 3 months to evaluate variation in pH at various storage conditions, i.e., 8 °C, 25 °C, 40 °C and 40 °C + 75% RH [34].

#### 2.2.19. Noninvasive Skin Investigation

Single blinded and split noninvasive techniques were performed for face studies, recommended for three months. Formulations for the left and right sides of cheeks were handed over to 11 volunteers for regular use for the study period after taking written consent. For this, Multi Skin Test Center MC 1000 (Courage + Khazaka Electronic GmbH, Cologne, Germany) with software Complete Skin Investigation (CSI) was used for the investigation of skin erythema, moisture, transepidermal water loss (TEWL), sebum level, pH, elasticity, pores size, wrinkles and skin pigmentation [41,42]. Ethical Standard Biosafety Committee COMSATS University Islamabad, Abbottabad Campus, Pakistan (reference CIIT/ATD/BSC/17-07) approved this study.

### 2.3. Statistical Analysis

The experimental data were collected at least in triplicate for each activity. The data were analyzed through *t*-test or one-way ANOVA and Tukey’s post hoc test with GraphPad Prism 8.0 (GraphPad Software, San Diego, CA, USA). The statistical significance for every experiment was considered significant below 0.05.

## 3. Results and Discussion

### 3.1. Preparation of Extract

Mango kernel and peel dried fine powder extracted with a 22% yield. The extraction process was repeated three times with fresh methanol (1:10) to extract the maximum phenolic compounds. Due to MeOH amphoteric nature, methanol gives a higher yield [43]. The method was thought to be capable of producing maximum yield.

### 3.2. Identification of Compounds in M. indica Peel and Kernel Extract

The compounds were identified in the M-Ext of *M. indica* of the peels and kernel. The GC-MS chromatogram is given in Figure 1 and Appendix A. The molecular weight, molecular formula, chemical name, and CAS number of the components were determined in accordance with NIST library (Table 1). The components succinic acid, cyclohexylmethyl geranyl heptacosanoic acid, 25-methyl-, methyl e succinic acid, cyclohexylmethyl geranyl dodecanoic acid 2,6,10,14-tetramethyl-7-(3-methylpent-4-tridecanoic acid, 12-methyl-, methyl est tetradecanoic acid chloroacetic acid, tetradecyl ester 14-methylpentadec-9-enoic acid methyl es tetradecanoic acid, 10,13-dimethyl-, met ethyl 14-methyl-hexadecanoate methyl 5,12-octadecadienoate 6-octadecenoic acid, and l-(+)-ascorbic acid 2,6-dihexadecanoate were screened in the GC-MS analysis. GC-MS was done to see whether this plant species included any specific chemical or collection of compounds that would support its recent and future application as herbal remedy in both commercial and traditional contexts. Additionally, it assists in selecting the perfect techniques for extracting these components. As a result, these data were examined in relation to any potential biological or therapeutic implications for aged skin, wrinkles, and oxidative stresses.

### 3.3. Atomic Absorption Spectroscopy

Heavy metals profiling is important in order to fabricate desirable antiaging drug-delivery systems [21]. Heavy metals including arsenic (As), chromium (Cr), cobalt (Co), cadmium (Cd), iron (Fe) and lead (Pb) were checked and results are given in Figure 2. It was noted that heavy metals were in lower concentrations than recommended levels [44,45,46]. The As was undetectable in M-Ext. As and its salts are banned in topical drug-delivery systems under EU legislation and in Canada and Germany [44], as they cause skin darkening, hair loss and cancer [47]. The concentration of Cr was 0.024 ppm in M-Ext. The US FDA limit for Cr as an impurity is up to 50 ppm [48]. Co was found 0.092 ppm in M-Ext. Co is a component of vitamin B12, an essential vitamin but contact poisoning occurs from constant contact causing irritation and rashes that go away slowly [46]. Pb concentration in M-Ext was 1.025 ppm. The US FDA recommends a maximum level of 20 ppm for lead as an impurity in cosmetics. Cd was 0.015 ppm in concentration in M-Ext. Cd limit is 0.03 to 0.10 ppm in cosmetic formulation, used as coloring pigment [49]. Fe is found 0.277 ppm in M-Ext. It is an essential mineral, but in high concentrations may cause serious health issues [49,50]. The data revealed that M-Ext is safe to use in topical formulations, as per the data, which have been consistent with those of the US FDA.

### 3.4. Effects on AOE

Antioxidant enzymes play a major role in oxidative stress, which accelerates the aging process in skin. The effect of M-Ext on antioxidant enzymes was evaluated using mouse peritoneal macrophages. The oxidative stress damages exacerbate skin pigmentation and aging, results in skin homogeneity complexion, wrinkling, sagging, dryness, and roughness [4]. The GSTs are involved in both the transport and biosynthesis of endogenous compounds and cellular defense mechanisms against xenobiotics and oxidative damage by catalyzing the conjugation of reduced glutathione through its cysteine thiol [5]. The extract markedly induced the glutathione S-transferase (GST) concentration compared to the negative control, as shown in Figure 3A. GSH plays critical roles in maintaining redox homeostasis [6]. M-Ext elevated the GSH concentration compared to the LPS-treated group (Figure 3B). GSH is found in the cytosol of cells (1–10 mM), playing a key role in regulation of oxidative stress by elimination of many ROS and these reactions are catalyzed by GSTs [6]. Catalase decomposes hydrogen peroxide to harmless products such as water and oxygen, it is used against many oxidative stress-related diseases as a therapeutic agent [7]. The extract showed significant increase in the catalase assay when compared to the negative control Figure 3C. The M-Ext markedly increases the SODs concentration compared to the negative control (Figure 3E). Similarly, the positive control treated with vitamin C also significantly induced the antioxidant enzymes, catalyzing the conversion of superoxides (O_2_^−^) to less reactive H_2_O_2_ [7]. The loss of SOD activity is linked to membrane lipid peroxidation, protein carbonylation, and DNA breakage [8]. Lipid peroxidation (POD) is a well-known mechanism, used as an indicator of oxidative stress in cells. The POD concentration was significantly increased in the negative control group treated with the LPS. However, the M-Ext markedly attenuated the concentration of the POD (GPx) compared to the LPS treated group as shown in the Figure 3D. The positive control also markedly reduced the concentration of POD. Overall results of M-Ext on enzymes which can successfully affect the oxidative stress suggests that M-Ext contain high antiaging activity and an excellent option for topical antiaging formulations.

### 3.5. Total Phenolic and Flavonoid Contents

Total phenolic contents (TPC) of M-Ext were 90 mg GAE/g of dry weight. The TPC also depends on geographical locations. 70 mg CE/g TFC were reported as dry weight. The observed values were comparatively higher as reported by sultana et al., for Pakistani varieties [51]. Flavonoids are the most important class of bioactive compound with antiaging activity. In green formulations they can be the versatile cosmeceuticals to treat early aging and oxidative stresses [52].

### 3.6. DPPH Assay

The free-radical scavenging activities of M-Ext were 83% at 150–200 μg/mL, as shown in Figure 4, IC_50_ of M-Ext was 23.35 μg/mL, and ascorbic acid showed 5.5 μg/mL IC_50_ free radical scavenging activities. The DPPH assay results indirectly prove the efficiency of the M-Ext against aging and oxidative stresses due to the presence of TPC and TFC. Outcomes based on DPPH activity of M-Ext, M-Ext is suitable as an antiaging or antioxidative stress agent in pharmaceutical or pharmaceutical formulations [14,53]. Therefore, it could be a new approach to formulate green formulation to cope with early aging and oxidative stresses that results in thwarted life.

### 3.7. Physicochemical Characterization of M-Ext-Loaded NLCs

M-NLCs showed 172.3 ± 14.7 nm average size (Figure 5A), 0.23 PDI and 90.7 ± 1.1% entrapment efficiency. B-NLC showed a 164.5 ± 8.6 nm average size (Figure 5B) with less than 0.3 PDI, indicating homogeneity. An increase in M-NLCs particle size was found, which may be due to the loading of mango peels and kernel phenolic-rich extract. The size distribution of M-NLC was also increased from 0.15 to 0.23. The PDI shows that particles were uniformly distributed [18]. Yichao et al. (2020) reported the blank rhamnolipid (Rh) nanoparticles with (118.7 nm diameter) quite comparable with the B-NLC (119.5 nm) [54]. Long Ba et al. (2016) also reported Rh-based nanoemulsion with around 130 nm size at room temperature [55], so the NLCs had excellent size distribution, which is thought to be a sign of success in nanodrug-delivery systems.

The Z-potential of blank NLCs was −77.7 ± 5.57 mV (Figure 5D). Rh produces negative zeta-potential [54], ascribed to its carboxylic groups [56,57]. Long Ba et al. (2016) reported Rh-based nanoemulsion with ZP −78 mV at room temperature [55]. The M-NLCs showed −64.5 ± 9.1 mV (Figure 5C) zeta-potential, i.e., a slight decrease in ZP was occurred. The produced M-NLCs have the desired ZP, i.e., less than −30 mV required for stable formulation [33]. The slight negative Z-potential is good for topical formulations, penetrating the skin.

### 3.8. Fourier-Transform Infrared Spectroscopy (FTIR)

FTIR examination is an operative technique for fast identification of loaded extract and determining interface between M-Ext and lipid matrix during the formulation procedure of M-NLCs. The FTIR assessments of B-NLCs, M-Ext, and M-NLCs are given in Figure 6A. The B-NLC spectrum showed its attributed peaks at 1050 cm^−1^ strong and narrow peak was attributed to S=O stretching), 1125 cm^−1^ (C-N stretching), 1425 cm^−1^ (C-H bending of alkane methyl group), and 1620 cm^−1^ (C=C stretching of conjugated alkene), 2875 cm^−1^ (peak was weak and slightly broad attributed to O-H stretching of alcohol intramolecular bonded), 2910 cm^−1^ medium peak showing C-H stretching alkane and at 3400 cm^−1^ broad peak showing alcohol O-H stretching. FTIR spectrum of M-Ext displayed characteristic peaks at 800 (C-H bending), 1050 cm^−1^ (CO-O-CO stretching), 1450 cm^−1^ (O-H bending), 1730 cm^−1^ (C=O stretching, α, β-unsaturated ester), 2875 cm^−1^ (C-H stretching, alkane), 2910 cm^−1^ (C-H stretching, alkane). FTIR spectrum of M-NLCs (M-Ext loaded NLCs) revealed characteristic peaks attributed at 700 cm^−1^, at 1050 cm^−1^ (CO-O-CO stretching anhydride), 1125 cm^−1^ (C-N stretching amine), 1425 cm^−1^ (C-H bending alkane methyl group), 1450 cm^−1^ (O-H bending carboxylic acid), 1620 cm^−1^ (medium, N-H bending), 1730 cm^−1^ (C=O stretching α,β-unsaturated ester), 2830 cm^−1^ (weak and slightly broad attributed to O-H stretching alcohol intramolecular bonded), 2910 cm^−1^ (C-H stretching alkane) cm^−1^, at 3400 cm^−1^ (O-H stretching alcohol intermolecular bonded). M-NLCs spectra confirm the M-Ext into NLCs and no chemical reaction between them. In the FTIR spectrum of M-NLCs, some characteristic bands of M-Ext were extinct in B-NLCs peaks, that shows the confirmation of M-Ext entrapment in M-NLCs [33].

### 3.9. Morphological Observation

The M-NLCs external morphology is given in Figure 6B. The M-NLCs dispersion in crystallized structure was observed via TEM [33]. The TEM images of M-NLCs are depicted as round shaped. The particles spherical morphology is beneficial for efficient delivery of M-Ext to the deep skin layers.

### 3.10. Cytotoxicity Studies

The viabilities of the HaCaT and fibroblast cells incubated with M-NLC and M-Ext for 48 h, at varied concentrations (0–250 µg/mL) are shown in Figure 7A,B. These data demonstrate that M-NLCs are safer as a topical pharmaceutical formulation and concentration-independent. Furthermore, M-Ext (free extract), at 10 µg/mL concentration in HaCaT cells, showed (Figure 7A) significant (* *p*) toxicity (Appendix A) compared to M-NLC, that was further increased with the increasing extract concentration. The same pattern was followed in the case of fibroblast cells (Figure 7B), where viability difference (* *p*) was found at 20 µg/mL (Appendix A). These results confirm the preferred choice of NLCs to boost the safe delivery of M-Ext.

### 3.11. Cell Permeation Studies

The HaCaT cells monolayer presents the same absorption assets as skin cells [49], the cells were grown on the transwell inserts and after 17 to 21 days of incubation, the monolayer was suitable for permeation experiments. The mass transport of M-NLCs and M-Ext was assessed across transwell inserts through cells monolayer after 1, 2, 3, and 6 h, as given in Figure 8. The permeation of M-Ext was significantly (** *p* < 0.001) less than M-NLCs (given in Appendix A). The results illustrate that the cell permeation of M-NLCs was higher than M-Ext. This significant increase in permeation defines the aptness of the prepared green NLCs for M-Ext as topical nanodelivery system. These outcomes provide the proof of concept for formulation green nanocarrier system for the delivery of natural antiaging and antioxidative agents for the possible clinical outlook.

### 3.12. Ex Vivo Diffusion Studies

The ex vivo release rate was determined via modified Franz diffusion using rat skin, shown in Figure 9A. M-Ext loaded O/W emulsion release above 80% of total PCs in 8 h, while M-NLC loaded emulsion (M-NLC-E) releases about 50% of PCs simultaneously. M-NLC-E showed slow release of PC compared to M-Ext loaded emulsion. Moreover, the release data of M-Ext from NLCs loaded into the emulsion and free M-Ext loaded emulsion via rat skin was analyzed through kinetic equations. PC release from carriers, with the *n* ≤ 0.43 shows that PCs release is controlled through Fickian diffusion mechanics, while *n* ≥ 0.85 is directed for dissolution process, and a mixture of 2 mechanisms is indicated by 0.43 < *n* < 0.8585 [58,59]. In this study, for M-NLC-E, the Higuchi model (R^2^ = 0.9764) was the best fitting equation shown in Figure 9B. This pattern is founded on idea that nutraceutical solubility is less than the early concentration in matrix, one dimensional diffusion of materials, matrix swelling is negligible and bioactive component diffusion is constant. The results were in better agreement with the earlier study of Aditya et al. who stated that NLCs presents a pronounced release at the initial stage followed by a sustained release [60].

### 3.13. Rheological Studies

Viscosity is often used to define the formulations profile for best delivery and allows formulators to optimize the personal care product [61]. Viscosities of M-NLC-E were found to be appropriate for skin application (criteria for skin application). Rheograms in Figure 10 show average viscosity 0.163 Pa·s with 0.026 standard deviation, which shows shear-thinning behavior reflecting pseudoplastic tendency. In rheology, deformation and flow of materials under force are studied [42], and formulations with the pseudoplastic flow make a film easily that covers the skin; that is why this is required for ideal topical preparations [62]. The presented rheological behavior of M-NLC-E showed smooth flow after applying tension that indicates high dispersion capacity during application. This property leads to easy creation of uniform film on skin surface.

### 3.14. pH of M-NLCs at Different Storage Conditions

pH is the quantitative measurement of solutions acidity or basicity. The change in pH can irritate the skin [63]. In the present study; the pH varied b/w 4.8–5.3 and showed no significant variation (given in Appendix A) at different storage conditions after 1st, 2nd and 3rd months as shown in Figure 11. The formulation should be stable in its entire shelf life. The buffer capacity of M-NLC-E maintained the pH under the influence of stressors nearly constant [40].

### 3.15. Noninvasive Skin Investigation

A noninvasive skin investigation was carried out on 11 volunteers for up to 3 months to evaluate the effect of M-NLC-E on skin erythema, melanin contents, moisture level, TEWL index, sebum contents, elasticity, wrinkles, pH, and skin pores.

#### 3.15.1. Erythema and Skin Melanin Level

The blank and loaded emulsions were applied on the left and right cheeks, respectively. The results are shown in Figure 12A. The M-NLC-E showed a significant reduction in erythema. The antioxidant compounds in M-Ext decrease free radical concentration in the skin [64,65]. The protective film of M-NLC-E on the skin also reduces the skin erythema by blocking pathogens, pollution, and UV radiation [66].

Melanins are the pigments at the dermo-epidermal junction, and the tyrosinase enzyme performs a critical part in its production [66]. In this study, Figure 12B shows that M-NLCs significantly decreased the amount of melanin after the 2nd and 3rd months. The M-Ext contains tyrosinase inhibitory activity [11,12] and is used for the treatment of melasma, freckles, and spots related to aging [67]. The results showed that M-NLC-E delivered M-Ext successfully in topical green formulation that makes the formulation to achieve extraordinary in vivo outcomes against aging related manifestos and oxidative stresses.

#### 3.15.2. Skin Hydration and TEWL Index

In the current study, the M-NLC-E showed a significant enhancement in hydration after 3 months of topical application comparatively to B-NLC-E, as shown in Figure 13A. In addition, the TEWL index significantly decreased, as shown in Figure 13B. The overall effect of M-NLC-E on TEWL and hydration level can be described as a vehicle (emulsion) that decreases water evaporation by establishing a thin layer on the stratum corneum [63]. In case of B-NLC-E these values were insignificant with a matter of time. The slight change by B-NLC-E in TEWL index is associated with an insulation effect that increased skin hydration level [68] and this occlusive effect depends significantly on the particle size, small size makes a dense film on skin surface [63]. Mango extract contain high amount of vitamin A, which increases the hyaluronic acid level as reported by Toshida et al., 2012 [69] and it may be the possible reason of increased in volunteers skin hydration by M-NLC-E.

#### 3.15.3. Sebum Level and Elasticity of Skin

Sebum is formed in the sebaceous glands and its overproduction leads to oily skin, larger pore sizes, acne, and pimples [70,71]. The current study results in Figure 14A indicates that the sebum level was significantly decreased in case of M-NLC-E and B-NLC-E showed a slight change, which may be due to the 5α-reductase inhibitory activity [72]. Skin elasticity is its ability to stretch and dependent highly on collagen level [73]. Figure 14B shows that M-NLC-E treated side showed significant results with respect to baseline visit vs. 3 M. This increase in skin elasticity is due to the M-Ext as it has a reversible inhibitory effect on elastase and collagenase [13]. The decrease in TEWL index and increase in skin hydration level (Figure 10) also enhanced the skin elasticity as described by Dobrev (2000) [68].

#### 3.15.4. Wrinkles and pH of the Skin

Wrinkle formation is a prominent effect of skin aging, secondary to actinic elastosis and prominent in sun-exposed areas [74,75]. Figure 15A and Appendix A, shows that up to 50% of wrinkles are decreased by M-NLC-E from M1 (base reading) to M3. The M-NLC-E effect of reducing wrinkles is the net effect of reduction in collagenase activity, TEWL reduction, and increase hydration [74,75,76].

Skin pH ranges 4–6 and its low pH is necessary for its normal physiology [77], in pathological conditions like dermatitis, acne it exceeds than normal range [78]. It was found that there was a slight decrease in pH Figure 15B. In terms of time, the skin pH values of B-NLC-E treated side and M-NLC-E at baseline visit versus 1 M; 2 M; and 3 M were determined to be insignificant using Tukey’s test. Study reveals that during study period the natural pH range was maintained. The acidic nature of the *stratum-corneum* prevents colonization of *Staphylococcus aureus*, *Candida albican*, and *Malassezia* [77,78].

#### 3.15.5. Pore Size

Skin pores are found at the top of hair follicles with small openings covering the body, as shown in Figure 16B and Appendix A. Genetics influence the pores size, but aging, sun, and high level of sebum also enlarge the pores size. In the present study, pores sizes were significantly reduced after 3 months of M-NLC-E application, as shown in Figure 16A. According to B.Y. Kim et al. (2011), the increase in skin elasticity also reduces pores size and number [79]. In present study the reduction of pores size may be due the increase in elasticity, hydration level and reduction in sebum.

## 4. Conclusions

Natural green formulations, especially for topical applications, are always favored due to their strong compatibility and minimal side effects. On the other side, early aging and oxidative stress is always a problem in the context of good looks. The current study evaluated the use of NLCs for natural herbal formulations and found excellent antiaging efficiency of *Mangifera indica* L. peels and kernel extracts loaded in NLCs. To the best of our knowledge, for the first time a green nanoformulation of the *Mangifera indica* L. peels and kernel extracts in NLCs with extraordinary safety profile, in vitro and ex vivo efficiencies has been achieved. The study might be a backbone for developing antiaging NLC-based green formulations and can be considered for clinical applications in the future. Secondly, green formulations are always preferred over synthetic formulations in the sense of the environment, compatibility, and low toxicity. Therefore, the current formulation may be helpful in relieving clinical disturbances regarding aging and oxidative stresses.

## Figures and Tables

**Figure 1 biomedicines-10-02266-f001:**
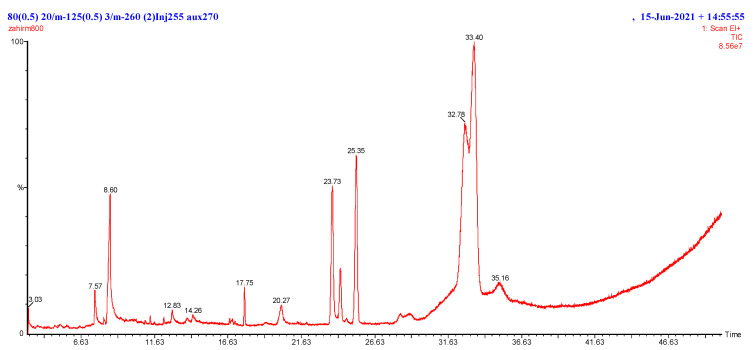
GCMS chromatogram of the M-Ext of peel and kernel extract.

**Figure 2 biomedicines-10-02266-f002:**
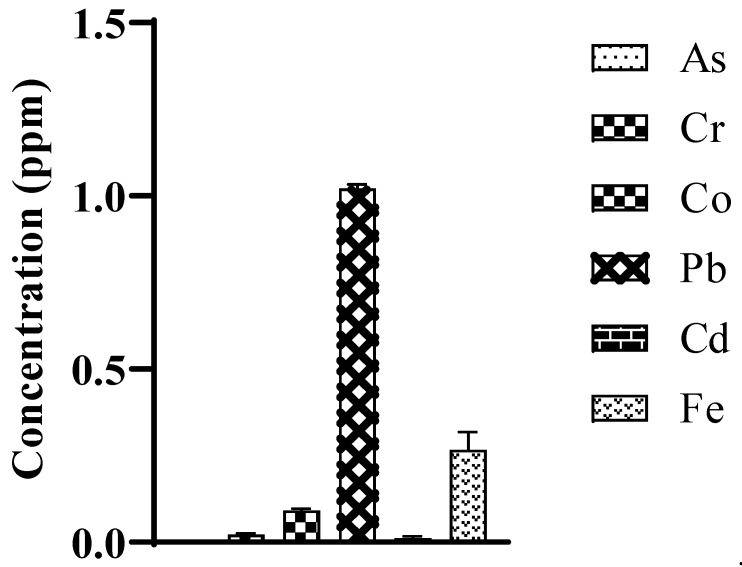
Heavy metal concentration (ppm) present in M-Ext analyzed through atomic absorption spectroscopy. Data was shown as means ± SD (*n* = 3).

**Figure 3 biomedicines-10-02266-f003:**
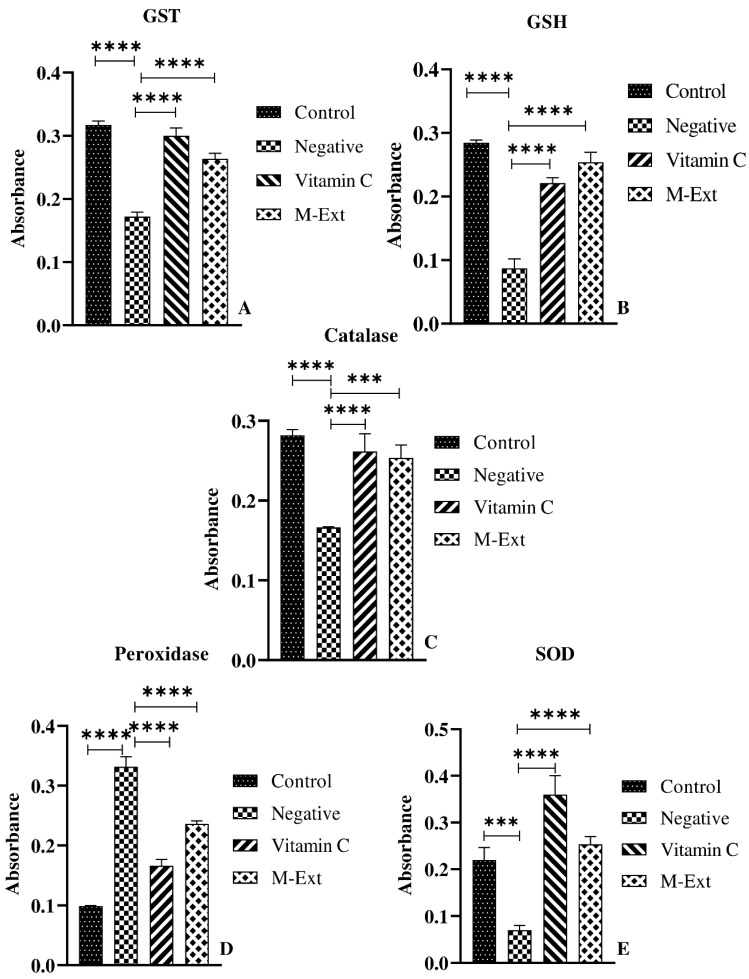
Effect of M-Ext on antioxidant enzymes (**A**) GST, (**B**) GSH, (**C**) catalase, (**D**) Peroxidase, and (**E**). Data shown as means ± SD (*n* = 3), *** *p* < 0.001, **** *p* < 0.0001.

**Figure 4 biomedicines-10-02266-f004:**
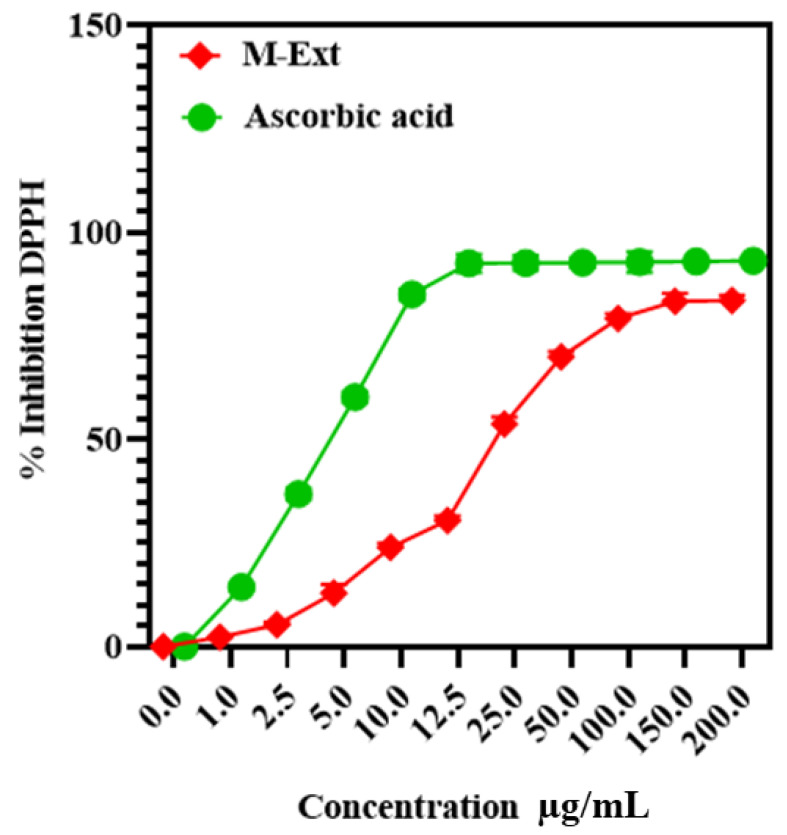
IC_50_ of ascorbic acid and M-Ext (extract of mango peels and kernel). Data shown as means ± SD (*n* = 3).

**Figure 5 biomedicines-10-02266-f005:**
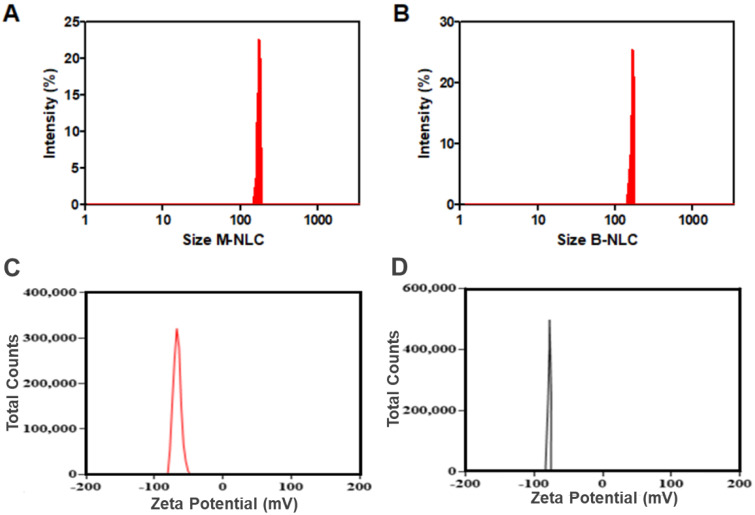
Particle size analysis. Size distribution of M-Ext-loaded NLCs (**A**) and blank NLCs (**B**). Zeta-potential of M-Ext loaded NLCs (**C**) and blank NLCs (**D**). Data shown as means ± SD (*n* = 3).

**Figure 6 biomedicines-10-02266-f006:**
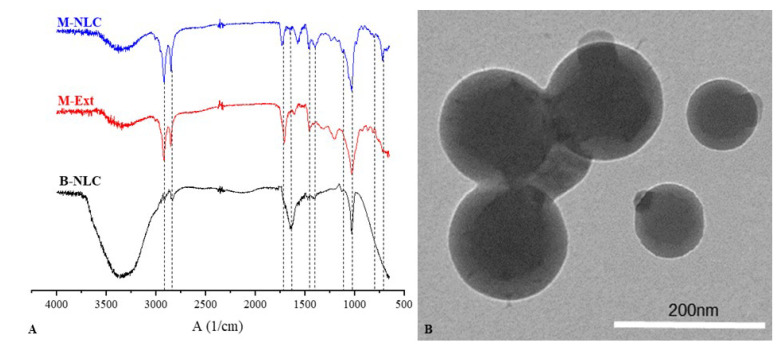
FTIR spectra of B-NLC; blank NLCs, M-Ext; *M. indica* peels and kernel extract, M-NLC; M-Ext loaded NLCs (**A**), TEM image of M-NLC (**B**).

**Figure 7 biomedicines-10-02266-f007:**
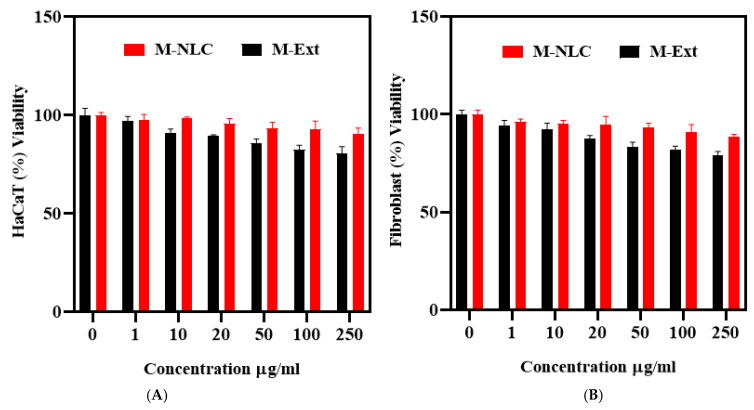
Cytotoxicity assay. Standard MTT assay was used to evaluate cytotoxicity of M-NLC and M-Ext on HaCaT cells (**A**) and Fibroblast cells (**B**). Data shown as means ± SD (*n* = 3).

**Figure 8 biomedicines-10-02266-f008:**
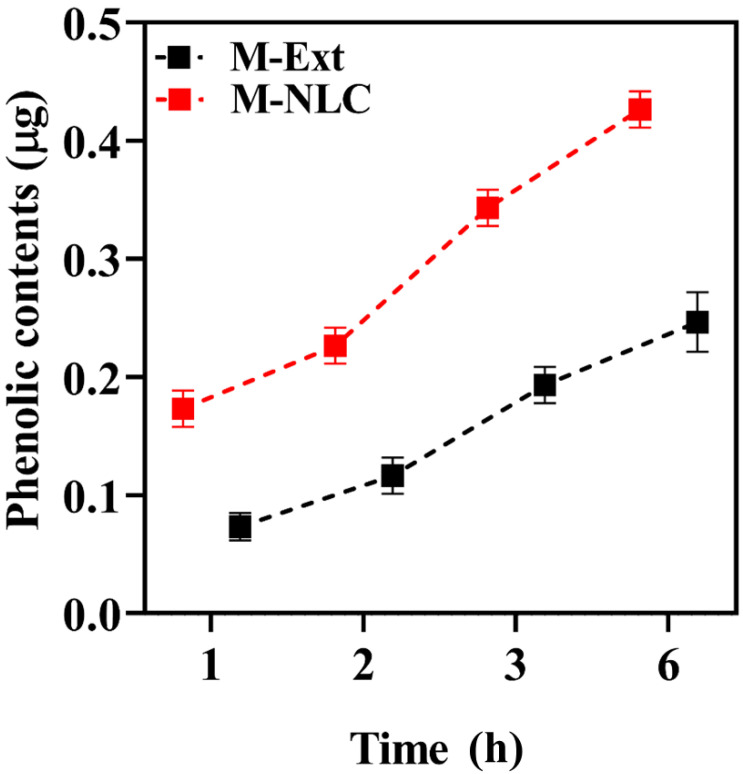
In vitro permeation study. HaCaT cells monolayer was used to determine the rate of permeation of M-NLC and M-Ext. Data shown as means ± SD (*n* = 3).

**Figure 9 biomedicines-10-02266-f009:**
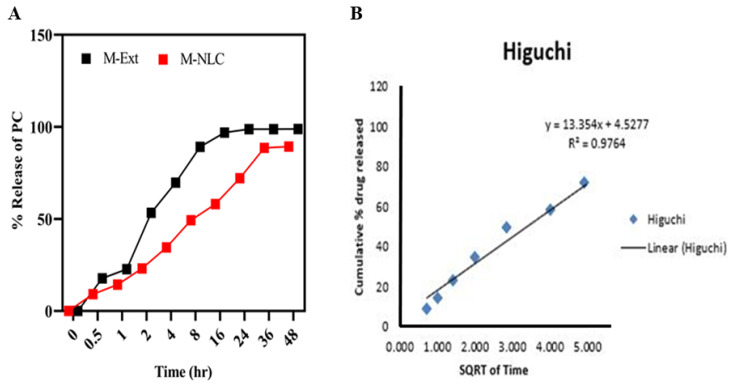
Ex vivo diffusion model. Ex vivo diffusion studies were performed on rat skin. (**A**) diffusion and release kinetics through rat skin. (**B**) Higuchi model. Data shown as means ± SD (*n* = 3).

**Figure 10 biomedicines-10-02266-f010:**
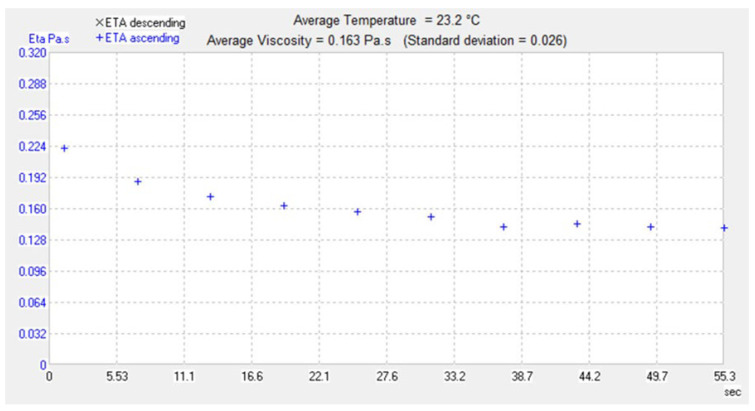
Rheological study. The rheogram showing flow behavior of the formulation.

**Figure 11 biomedicines-10-02266-f011:**
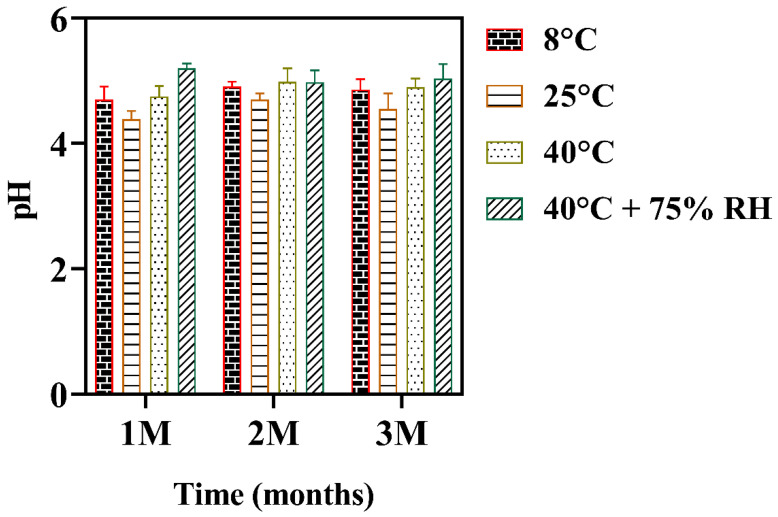
pH changes at various storage conditions. Data shown as means ± SD (*n* = 3).

**Figure 12 biomedicines-10-02266-f012:**
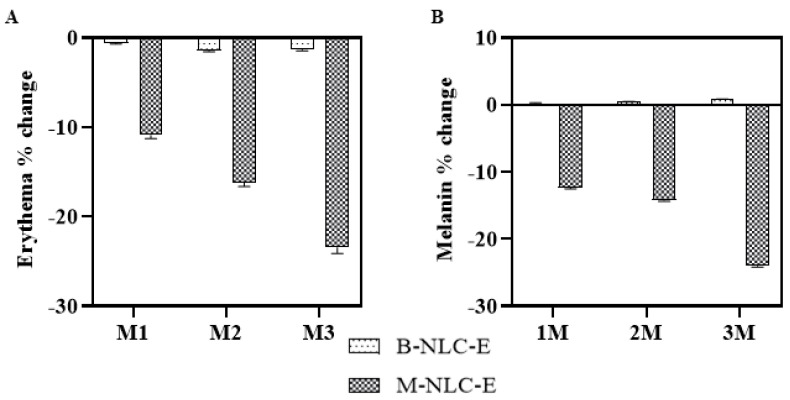
Erythema (**A**) and Melanin contents (**B**) change after 1st (M1), 2nd (M2) and 3rd month (M3) of use of B-NLC-E and M-NLC-E.

**Figure 13 biomedicines-10-02266-f013:**
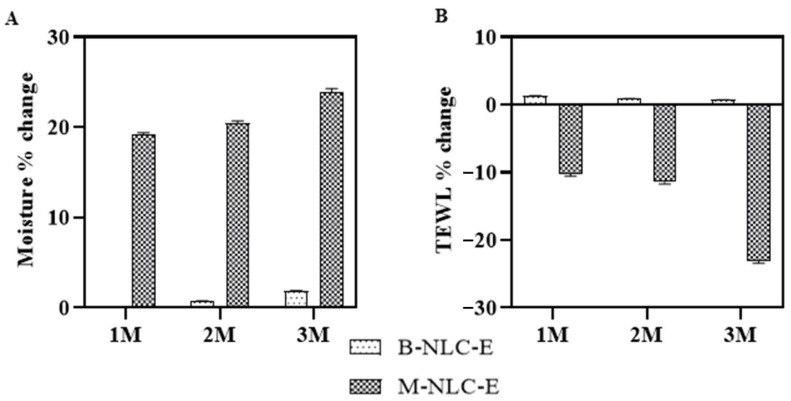
The moisture changes (**A**) and TEWL loss (**B**) after 1st (M1), 2nd (M2) and 3rd month (M3) of use of B-NLC-E and M-NLC-E.

**Figure 14 biomedicines-10-02266-f014:**
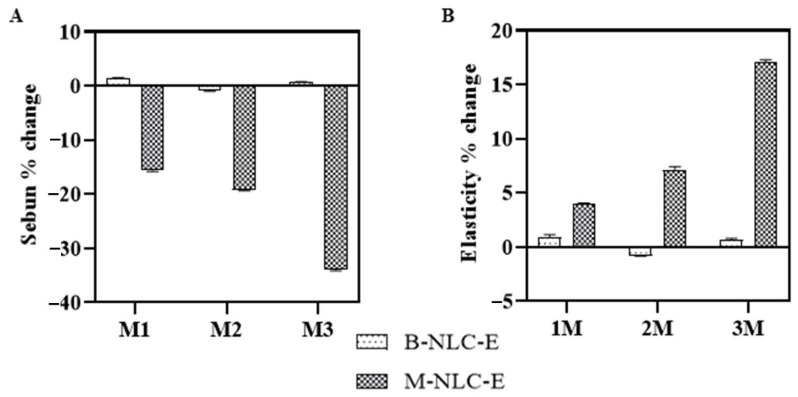
Analyses of skin sebum (**A**) and elasticity (**B**) after 1st (M1), 2nd (M2) and 3rd month (M3) of use of B-NLC-E and M-NLC-E.

**Figure 15 biomedicines-10-02266-f015:**
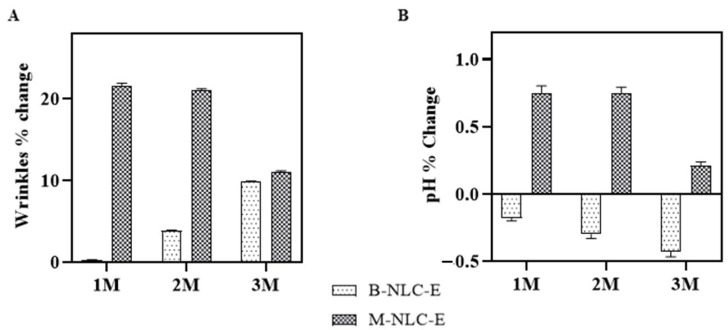
Wrinkles (**A**) and skin pH (**B**) analysis after 1st (M1), 2nd (M2) and 3rd month (M3) of use of B-NLC-E and M-NLC-E.

**Figure 16 biomedicines-10-02266-f016:**
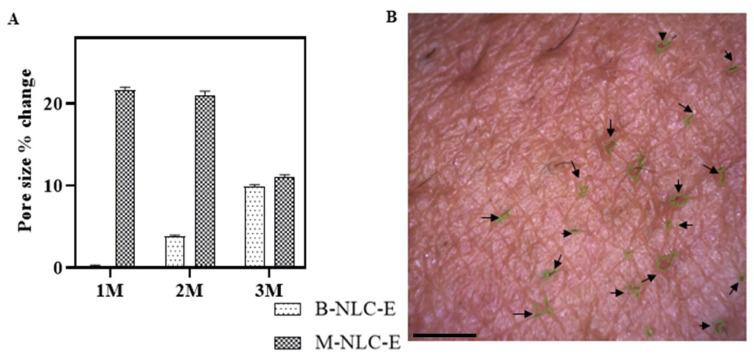
Pore sizes of the skin (**A**) after 1st (M1), 2nd (M2) and 3rd month (M3) of use of B-NLC-E and M-NLC-E (**B**) Pore size after 3rd month. Scale bar is 1 µm. Images were taken through Visioscan attached with CSI. Data shown as means ± SD.

**Table 1 biomedicines-10-02266-t001:** Identification of compounds in the peel and kernel M-Ext through GCMS analysis.

Hit	Rev	For	Compound Name	M.W.	Formula	CAS	Library
1	753	517	Succinic acid, cyclohexylmethyl geranyl	350	C_21_H_34_O_4_	900391-21-5	NIST
1	878	561	Heptacosanoic acid, 25-methyl-, methyl e	438	C_29_H_58_O_2_	900112-14-5	NIST
1	819	704	Dodecanoic acid	200	C_12_H_24_O_2_	143-07-7	NIST
1	803	533	2,6,10,14-tetramethyl-7-(3-methylpent-4-	348	C_25_H_48_	900370-41-6	NIST
1	943	882	Tridecanoic acid, 12-methyl-, methyl est	242	C_15_H_30_O_2_	5129-58-8	NIST
1	858	774	Tetradecanoic acid	228	C_14_H_28_O_2_	544-63-8	NIST
1	894	812	14-methylpentadec-9-enoic acid methyl es	268	C_17_H_32_O_2_	900365-89-7	NIST
1	933	661	Tetradecanoic acid, 10,13-dimethyl-, met	270	C_17_H_34_O_2_	267650-23-7	NIST
1	869	754	Ethyl 14-methyl-hexadecanoate	298	C_19_H_38_O_2_	900336-64-7	NIST
1	877	735	Methyl 5,12-octadecadienoate	294	C_19_H_34_O_2_	900336-43-1	NIST
1	936	815	6-octadecenoic acid	282	C_18_H_34_O_2_	900336-66-8	NIST
1	811	683	L-(+)-ascorbic acid 2,6-dihexadecanoate	652	C_38_H_68_O_8_	28474-90-0	NIST

## Data Availability

Data are contained within the article.

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
