# Peer review of "In Vitro and Ex Vivo Evaluation of Mangifera indica L. Extract-Loaded Green Nanoparticles in Topical Emulsion against Oxidative Stress and Aging"

_biomedicines, 2022, doi:10.3390/biomedicines10092266_

Round 1

Reviewer 1 Report

The article “In vitro and Ex vivo Evaluation of the Mangifera indica L. Extract Loaded Green Nanoparticles in Topical Emulsion against Oxidative Stress and Ageing” is very well written and a promising effort has been made to cope with ageing and oxidative stresses in a natural way by formulating green nanoparticles. I recommend the paper for publication. However, few answers of the following questions should be clear before publication.
1.    The paper should be thoroughly reviewed by English native speaker for grammatical mistakes.
2.    Why the nanoparticles were loaded into an emulsion?
3.    Figure 5, the figures quality is very poor, needs to improve.
4.    Why HaCaT cells and Fibroblast cells were choosed for cytotoxicity studies?
5.    Figure 5, the figures quality is very poor, needs to improve.
6.    There is no control in Erythema and skin melanin level.
7.    Figure 1, at page 30 should be corrected as Figure 13.
8.    Figure 2, at page 31 should be corrected as Figure 14.
9.    There is no underlying molecular mechanism involved in ageing here, it can be included?

Author Response

We want to express our humble gratitude and appreciation for kindly reviewing our article entitled “In vitro and Ex vivo Evaluation of the Mangifera indica L. Extract Loaded Green Nanoparticles in Topical Emulsion against Oxidative Stress and Ageing”, pointing out mistakes, and giving constructive comments concerning our manuscript. We appreciate the time and efforts that you dedicated to providing feedback on our manuscript and grateful for the insightful comments and valuable improvements to our manuscript. We thoroughly considered and followed all suggestions and criticisms.

All issues raised in comments have been addressed to substantially promote the quality of this manuscript. All the concerns and questions were answered point by point in this letter and the change in the manuscript has been highlighted.

Please accept our sincerely appreciation for having pointed out the questions and hope this letter will make ourself understood. Thank you very much.

Reviewer 2 Report

The chemical analysis over the extract is not the appropriate. The evaluation of the chemical composition of methanol Lucas today contains some important fault. If you look for the chemical composition you’ll find molecules that are not natural ex fluoro acetic acid esters. Since you have not any clear term Dunkel position how you can defend the activity of your extract

Author Response

(The authors gave the same response as above.)
